# Differing Roles of Bacterial and Fungal Communities in Cotton Fields by Growth Stage

**Dongwei Li [1], Yuhui Yang [2], Yulong Zhao [1], Guangli Tian [1], Mingsi Li [3], Husen Qiu [4],\* and Xinguo Zhou [1]**

1 Farmland Irrigation Research Institute, Chinese Academy of Agricultural Sciences, Xinxiang 453002, China; lidongwei@caas.cn (D.L.); zhaoyulong@caas.cn (Y.Z.); tianguangli@caas.cn (G.T.); zhouxinguo@caas.cn (X.Z.)
2 College of Water Resource and Architectural Engineering, Tarim University, Alaer 843300, China; 120130022@taru.edu.cn
3 College of Water Conservancy and Architecture Engineering, Shihezi University, Shihezi 832002, China; limingsi@shzu.edu.cn
4 School of Environment and Surveying Engineering, Suzhou University, Suzhou 234100, China
\* Correspondence: qiuhusen@ahszu.edu.cn; Tel.: +86-18568219098

**Abstract:** The global demand for cotton makes sustainable cotton production an important issue that can be improved by a better understanding of the influence of soil microbes on cotton growth. We collected cotton field soils at the seedling and flowering/boll-setting (FBS) stages in order to obtain soil properties and cotton growth indices. Bacterial and fungal community compositions were assessed by high-throughput sequencing of 16S rRNA and internal transcribed spacer genes, respectively, after which the differences in microbial functions and their influencing factors at different growth stages were analyzed. Both the diversity and composition of soil bacterial and fungal communities were found to be significantly different between the seedling and FBS stages. Microbes in the seedling stage had significantly higher richness and biomass than those in the FBS stage. Compared with the seedling stage, the stability of the soil bacterial communities was decreased. The cotton growth indices at both the seedling and FBS stages were associated with compositional shifts in the bacterial community and but not the fungal community. The abundance of specific soil microbial taxa (e.g., Pseudarthrobacter, Thiobacillus, Cephalotrichum, Chaetomium, and Fusarium) were correlated with cotton growth indices at the seedling stage, being mainly regulated by soil salinity and nitrate content. Our results highlight the importance of soil microbial communities in mediating cotton growth and will be useful in providing better strategies for the improvement of cotton agriculture.

**Keywords:** agriculture soils; cotton growth stages; bacterial community; fungal community; cotton growth indices

## 1. Introduction

More than 30% of the irrigated lands in arid and semi-arid areas are affected by soil salinity [1], severely restricting sustainable economic development in arid areas. Xinjiang, in China's inland northwest, has extremely arid climate conditions that make it a prime region for soil salinization and secondary salinization. Cotton is a major cash crop in the region, being produced by nearly half of local farmers and accounting for >38% of their net income [2]. Cotton growth in high-salinity environments is a major challenge as its seedlings are very sensitive to salt stress [3,4], and this stage determines their establishment and subsequent later growth [5,6]. Addressing this is an important aspect of achieving the sustainable utilization of saline alkali land.

Microbial communities play an important role in supporting ecosystem stability and function [7]. In agricultural ecosystems, soil microbial communities drive the biogeochemical cycles of critical elements and maintain soil productivity [8], and the relationships between soil microbial communities and crop growth and health have been extensively documented [9,10]. Microbial diversity in agricultural soils varies with environmental

gradients and nutrient availability [11,12], while differences in crop species' nutrient uptake and environmental stress adaptations also lead to heterogeneity in bacterial growth and abundance [13]. These factors can result in an inhomogeneous distribution of soil microbial communities throughout the crop growth period in an agricultural field [14]. Studies have shown that soil microbial communities associated with a wide range of plants (e.g., Arabidopsis, Medicago, corn, pea, wheat, and sugar beet) changed significantly during plant development [14]. It is therefore reasonable to speculate that soil microbial communities could vary between different cotton growth stages. An improved understanding of these dynamics during cotton growth could greatly contribute to promoting nutrient cycling and enhancing nutrient utilization efficiency.

Soil microbial communities (bacteria, fungi, etc.) represent complex environments with high biodiversity [15] and versatile physiologies that help regulate many soil ecological processes [16]. Different biological populations exhibit fine differences in body size, metabolic activity, and susceptibility to environmental changes [17,18]. Many bacteria in soil environments depend on fungal hydrolysis of materials to form primary substrates [19], and in return contribute to resources needed for fungal decomposition by providing nutritional benefits [20]. Bacteria can also consume fungal exudates and biomass to promote carbon and nitrogen cycling [21]. In addition, fungi and bacteria existing in the same place are known to dynamically compete for space, nutrients, and essential minerals [22,23]. The physical and chemical properties of soil can regulate the biomass, diversity, and composition of soil microbial communities in agricultural ecosystems [24–26]. Although various studies have investigated soil microbial communities in this context, the distribution and dynamics of soil bacterial and fungal communities during cotton cultivation and the environmental factors regulating them remain unclear.

In this study, we used subsurface drainage treatments with soils from a salinized cotton field to characterize the effects of cotton growth stage on the diversity and composition of bacterial and fungal communities and their interactions with cotton growth and soil properties. We aimed to (i) observe any differences in the diversity, biomass, and composition of soil bacterial and fungal communities between the seedling and flowering/boll-setting (FBS) stages and (ii) explore how soil microbes affected cotton growth and their potential regulatory factors. We performed high-throughput sequencing based on 16S rRNA and internal transcribed spacer (ITS) genes to identify the bacterial and fungal communities. We then used statistical differences and correlation analyses to distinguish the stage-related differences in microbial communities and determine how environmental changes in soils led to variations in microbial communities and potential effects on cotton growth. Our results provide novel insights into soil microbial functions in cotton agriculture.

## 2. Materials and Methods

### 2.1. Experiment Description and Sample Collection

The experiment was performed from April to October 2019, in the 16th Regiment of Aral City, 1st Division, Xinjiang (80°30′–81°58′ E, 40°22′–40°57′ N) (Figure 1). This site has an altitude of 1025 m, an average annual precipitation of 40.1–82.5 mm, and an average annual evaporation of 1976.6–2558.9 mm. During the experimental period, the average temperature was 22.3 °C, and the average sunshine duration was 8.3 $h \cdot d^{-1}$. This region belongs to a dry climate zone, and groundwater is found at 0.6–1.0 m depth. The test field soil was mainly composed of sandy loam with a dry bulk density of 1.66 $g \cdot cm^{-3}$, while the field water capacity was 34.95% (volume ratio) and the saturated moisture content was 42.55%.

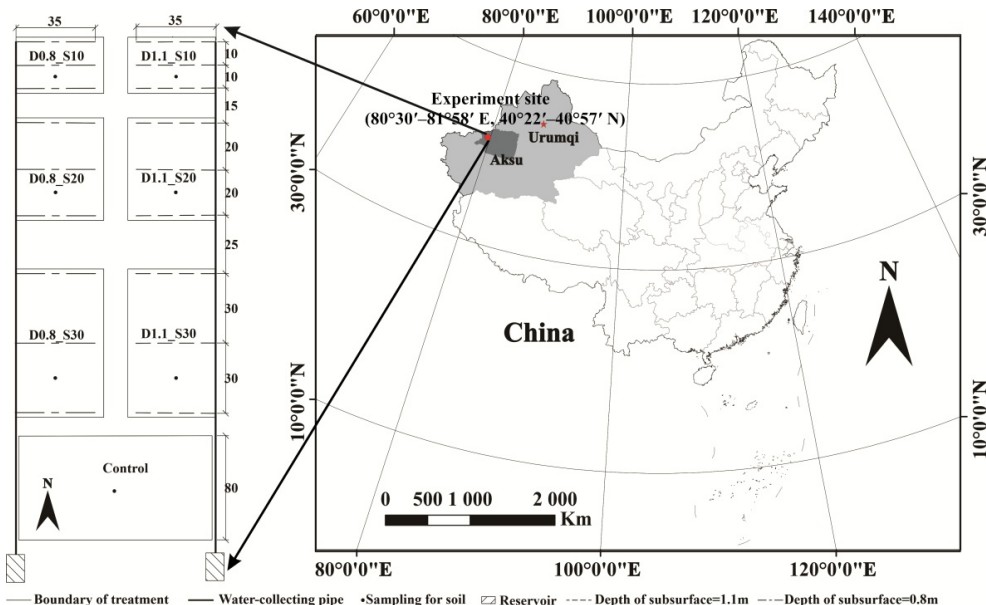

**Figure 1.** Schematic diagram of the geographic location of the experimental field and the subsurface drainage pipe layout of each treatment.

Subsurface water-absorption piping was installed in selected representative areas after the seed cotton harvest in October 2017, using polyvinyl chloride corrugated pipes with an inner diameter of 110 mm and a perforated surface (aperture ratio of 2.5%). Two layers of non-woven fabric were wrapped around the pipes, which were then connected to collection pipes of 110 mm inner diameter at the end. The gradient ratios of the absorption and collection pipes were 2‰ and 3‰, respectively. The collection pipes were connected to reservoirs. A small pumping station was used for drainage, operated whenever water was present. The treatments assessed two drainage depths (D: 0.8 and 1.1 m) and three drainage pipe spacings (S: 10, 20, and 30 m), as well as an undrained control (Figure 1), generating seven field trials: control (CK), D1.1_S10, D1.1_S20, D1.1_S30, D0.8_S10, D0.8_S20, and D0.8_S30.

All experimental fields were planted with the cotton variety Tamian No. 2. Seeds were sown in April 2019 with a wide and narrow row planting pattern (row spacing of 11 cm + 66 cm + 11 cm + 66 cm + 11 cm + 11 cm) and a plant spacing of 11 cm. The same fertilization level was used in each treatment during cotton growth, consisting of pure nitrogen, $P_2O_5$, and $K_2O$ at 350, 150, and 80 kg·hm$^{-2}$, respectively. Surface soil samples were collected at the seedling stage (28 May) and FBS (2 September). A multipoint sampling method was used to collect mixed soil samples at three random points for each treatment, with three samples being taken at each point. Part of the soil was taken as fresh soil samples and quickly stored in a refrigerator at 4 °C for the determination of soil physical and chemical indexes, while the other part was cryopreserved at −80 °C after rapid liquid nitrogen cooling for the determination and analysis of soil microbial communities.

## 2.2. Soil Properties and Cotton Growth Indices

The soil volume mass was determined using the cutting ring method. Soil moisture content was determined using the oven-drying method. Briefly, a dry soil sample was crushed into pieces and filtered using a 1 mm sieve. Next, 20 g of soil were placed in an Erlenmeyer flask, and 100 mL of distilled water was added. Samples were oscillated for 30 min, filtered, and left to rest for 10 min to obtain the leaching solution (water: soil ratio of 5:1). The conductivity of the leaching solution was determined using an electrical conductivity meter (DDSJ-308-a), and the soil salt content was determined via the drying

residual method using the following equation, which fitted the relationship between soil salinity and conductivity:

$$Y = 0.0044EC + 0.5438 \qquad R^2 = 0.9924 \qquad (1)$$

where Y represents the soil salinity ($g \cdot kg^{-1}$) and EC represents the conductivity ($\mu s \cdot cm^{-1}$).

Ten grams of fresh soil samples were placed in a 50 mL trigonometric bottle, and 50 mL KCl solution (superior pure GR) with a concentration of $2 \; mol \cdot L^{-1}$ was added. The mixture was oscillated for 15 min at $200 \; r \cdot min^{-1}$) at room temperature. The supernatant was extracted for soil $NH_4^+$ and $NO_3^-$ analysis using an AA3-HR continuous flow analyzer (Seal Analytical, Hamburg, Germany). Soil pH was measured in a 1:5 soil/water suspension using a pH meter. OM and AP concentrations were analyzed using a UV-8000 spectrophotometer after oxidizing organic carbon with potassium dichromate ($K_2Cr_2O_7$) and the molybdenum antimony colorimetric method, respectively.

Based on the emergence of seedlings in the field, five plots (2 m long and 1 m wide under the mulching film) in each treatment were random selected at 20 d post-sowing to calculate the seedling emergence rate. For each treatment, nine plants with uniform and representative growth were selected, and their plant height and leaf area were measured with a straightedge for each growth stage. At the harvest stage, plots with representative flowering status were selected to count the number of plants and the number of bolls per plant, followed by sampling of 20 seed bolls with uniform growth to calculate the mean boll weight. Finally, all cotton in each plot was picked and weighed to calculate the actual yield of seed cotton.

*2.3. Sequencing of Soil Microbial Communities*

The FastDNA® SPIN Kit for Soil (QIAGEN, Valencia, CA, USA) was used to extract total microbial DNA from each soil sample. DNA concentration and purity were detected using a NanoDrop 2000, and DNA extraction quality was detected by 2% agarose gel electrophoresis. When sequencing primers were designed, barcode sequences were added to the sequences to distinguish the sequencing data of each sample. The V3–V4 regions of bacterial 16S rRNA and ITS genes from each sample were amplified using primers 341F-806R (341F: ACTCCTACGGGAGGCAGCAG, 806R: GGACTACHVGGGTWTCTAAT) and ITS1-ITS4 (ITS1: TCCGTAGGTGAACCTGCGG, ITS4: TCCTCCGCTTATTGATATGC), respectively [27,28]. The amplification procedure was as follows: pre-denaturation at 95 °C for 3 min; 27 cycles of (denaturation at 95 °C for 30 s, annealing at 55 °C for 30 s, and extension at 72 °C for 30 s), and extension at 72 °C for 10 min (PCR instrument: ABI Gene Amp® type 9700). The amplification system was based on 20 μL, 4 μL 5× Fast Pfu buffer, 2 μL 2.5 m D NTPs, 0.8 μL primer (5 μm), 0.4 μL Fast Pfu polymerase, and 10 ng DNA template. PCR products were recovered using 2% agarose gel, purified using the Axy Prep DNA Gel Extraction Kit (Axygen Biosciences, Union City, CA, USA), eluted by Tris-HCl, and detected by 2% agarose electrophoresis. QuantiFluor™-ST (Promega, Madison, WI, USA) was used for the quantitative detection. PCR products were used to construct the sequencing library based on the quantitative results and sequencing requirements. The constructed libraries were sequenced on the Illumina MiSeq PE300 sequencing platform.

We removed reads with an average phred score < 20, ambiguous bases, homopolymer runs > 6, mismatches in primers, and sequence lengths < 150 bp [29]. The remaining high-quality reads were assigned to the samples based on their unique barcodes combined with the end of reverse primers. Subsequently, reads with an overlap > 10 bp and without any mismatch were assembled into tags using FLASH [30]. Tags with ≥97% similarity were assigned to the same operational taxonomic unit (OTU) using the QIIME v1.9.2 [31] software package. Representative sequences of each OTU were picked using the default method and assigned to a bacterial and fungal taxonomy based on the SILVA database [32]. The bacterial and fungal OTU abundance tables were constructed and normalized using a standard number of tags according to the sample with the lowest tag number.

### 2.4. Statistical Analyses

Alpha diversity indices of soil microbial communities, including Chao1 and Shannon, were calculated using QIIME v1.9.2 [33]. Boxplots based on the alpha diversity indices and soil biomass were developed using R v4.0.2, and differences between the seedling and FBS stages were analyzed using t-tests. Variations in the microbial community compositions of soils between the seedling and FBS stages were evaluated by principal coordinate analysis (PCoA) and the PERMANOVA test using the "vegan" package in R v4.0.2. The relative abundance of bacteria and fungi at the phylum level in the seedling and FBS stages was visualized using pie charts. STAMP was applied to recognize the significant different microorganisms in the soil microbial communities between the seedling and FBS stages [34]. The correlation (R) and significance (P) matrices were calculated using the Hmsic package in R version 3.6.2, and only strong correlations (Spearman's rank correlation coefficient, r > 0.9 (or r < −0.9) were selected for further analysis [35]. The constructed correlation matrix was transformed in Gephi version 0.9.3 to generate valid co-occurrence networks. A Mantel test was conducted to assess the correlations between bacterial and fungal community structure and environmental variables using the ecodist package [36]. Correlations between soil microbes, soil properties, and cotton growth indices were determined by the Spearman coefficient and the results were visualized by the pheatmap package in R v4.0.2.

### 3. Results

#### 3.1. Biomass and Alpha Diversity of Soil Bacterial and Fungal Communities

The bacterial and fungal biomasses were compared between the soils at the seedling and FBS stages using qPCR on the basis of the 16S rRNA and 18S rRNA genes, respectively. Compared to the seedling stage, both biomasses significantly decreased in soils at the FBS stage (*t*-test, $p < 0.05$, Figure 2a,b). Comparative analysis of the microbial communities between the seedling and FBS stages was performed by sequencing the relative gene amplicons. Soil bacteria had 11,149 OTUs with 97% similarity, clustered into 55 phyla, 191 classes, 465 orders, 764 families, and 1430 genera, while soil fungi had only 1240 OTUs, clustered into 11 phyla, 32 classes, 69 orders, 142 families, and 237 genera. The Chao1 index for both soil bacterial and fungal communities in the seedling stage was significantly higher than in the FBS stage (*t*-test, $p < 0.05$, Figure 2c,d). Significant differences in the Shannon index were also found in the soil bacterial communities between the seedling and FBS stages (*t*-test, $p < 0.05$, Figure 2c), but there was no significant difference in the Shannon index of soil fungal communities (*t*-test, $p > 0.05$, Figure 2d).

#### 3.2. Beta Diversity of Soil Bacterial and Fungal Communities

Both soil bacterial and fungal communities in the seedling and FBS stages were clearly separated into two distinct clusters by PCoA (Figure 3). The first two PCs accounted for 36% and 32% of the total variation in bacterial and fungal communities. The soil fungal communities were mainly classified by PC1, which explained 20% of the total variation (Figure 3b). In contrast, the soil bacterial communities from the seedling and FBS stages were distinguished by PC2, which only explained 12% of the total variation (Figure 3a). Moreover, PERMANOVA tests demonstrated that the growth stage had a significant impact on both the soil bacterial and fungal communities ($p < 0.05$), explaining 10.8% and 14% of the total variation, respectively. Overall, the effect of growth stage on soil fungal communities was more obvious than on bacterial communities.

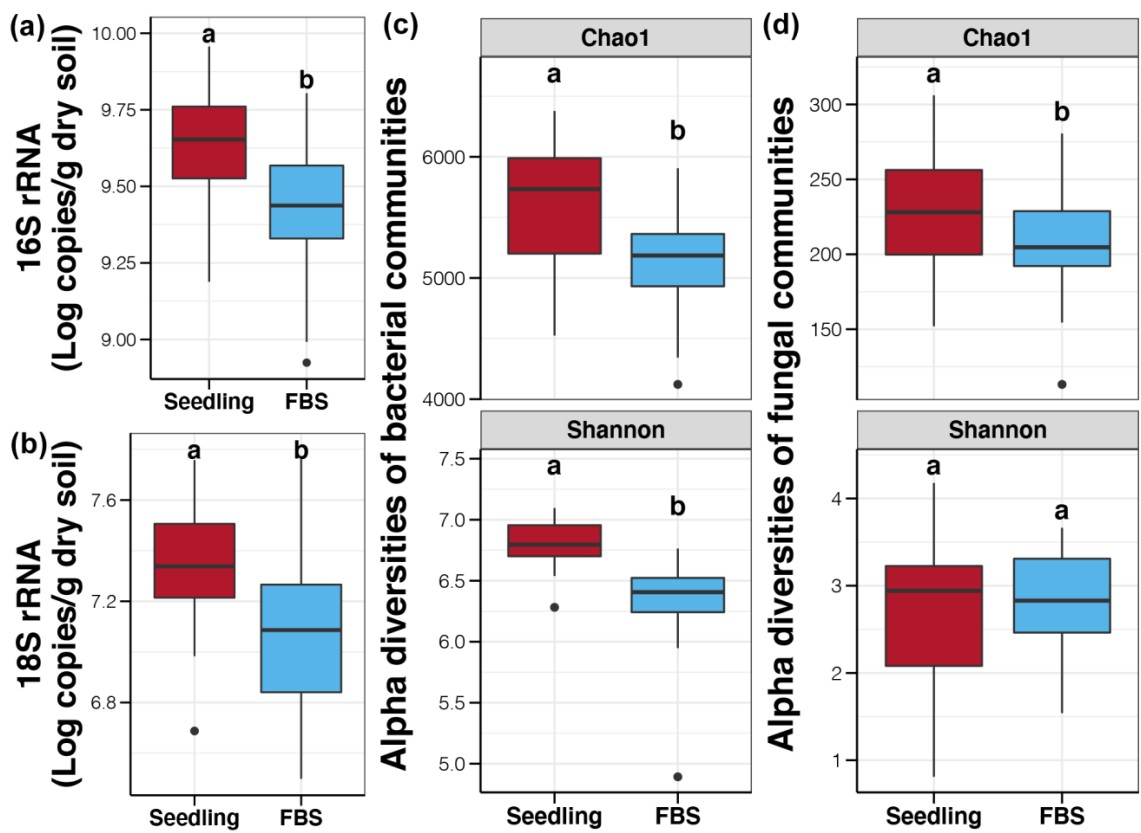

**Figure 2.** Comparison of the biomass and alpha diversity of microbial communities in cotton planting soils between the seedling and FBS stages. (**a**) Number of 16S gene copies in soil samples. (**b**) Number of 18S gene copies in soil samples. (**c**) Plots of Chao1 and Shannon indices of bacterial communities at the OTU level. (**d**) Plots of Chao1 and Shannon indices of fungal communities at the OTU level. Different lowercases letters above each box in the same subfigure represent significant differences between groups (*t*-test, $p < 0.05$).

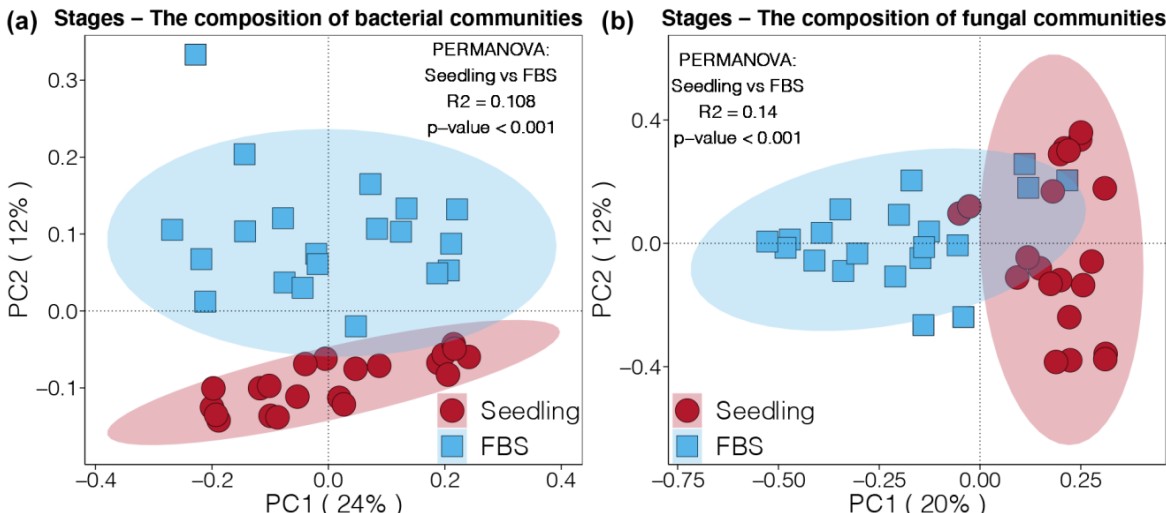

**Figure 3.** Principal coordinate analysis (PCoA) and PERMANOVA test of the soil bacterial (**a**) and fungal (**b**) communities in the cotton planting fields. The distance between the samples was calculated based on dissimilarity in OTU composition using the Bray–Curtis dissimilarity index. A greater distance between two points represents a higher dissimilarity between them.

### 3.3. Difference in Soil Bacterial Abundance between the Seedling and FBS Stages

Of the fifty-five soil bacteria phyla observed in different samples, Proteobacteria was the most dominant (followed by Firmicutes, Actinobacteria, Chloroflexi, and Acidobacteria), accounting for 74.96% and 77.05% in samples from the seedling and FBS stages, respectively (Figure 4a). The effect of growth stage on bacterial phyla was assessed using a Wilcoxon rank-sum test with FDR adjustment (Figure 4b). Proteobacteria, Nitrospirota, and Entotheonellaeota were more abundant in soils from the seedling stage, while Firmicutes was significantly enriched in soils from the FBS stage ($p < 0.05$). At the genus level, the relative abundances of Paenisporosarcina and Planococcus were significantly higher in soils from the FBS stage ($p < 0.05$, Figure 4c). In contrast, diverse bacterial genera were more abundant in soils from the seedling stage, including Sphingomonas, Gaiella, Bauldia, Lysobacter, Fictibacillus, and Thiobacillus ($p < 0.05$, Figure 4c).

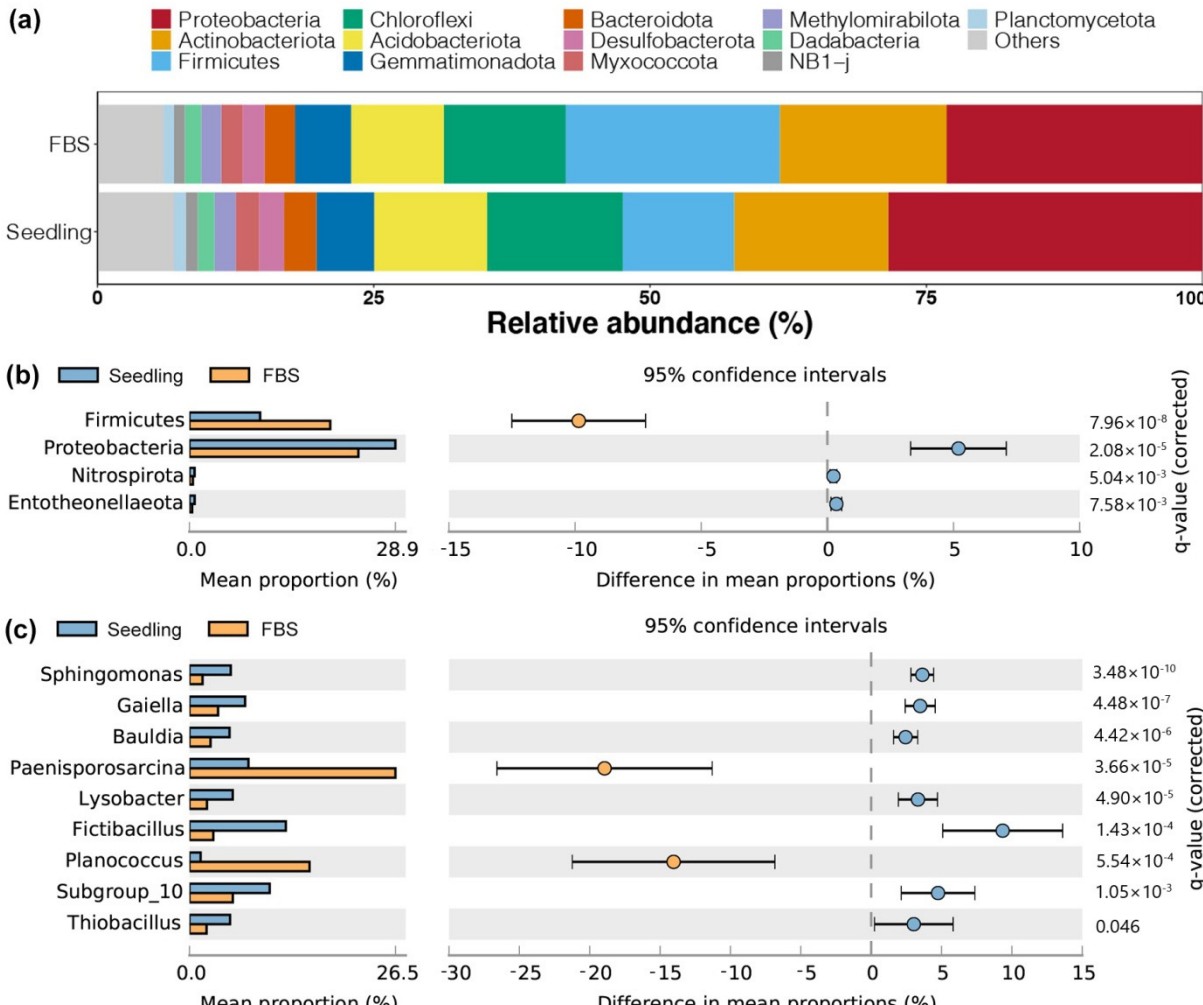

**Figure 4.** The average relative abundance of soil bacterial phyla in cotton planting fields at the seedling and FBS stages (**a**). Inter-group difference test of relative abundance of bacteria in soils between the seedling and FBS stages at the phylum (**b**) and genus (**c**) levels, respectively. If the q-value corrected value < 0.05 (*t*-test *p*-value with the FDR adjust), the phyla or genera were identified to be different between the seedling and FBS stages. The barplot on the left represents the average abundance of different bacteria in the seedling and FBS stages. The point plot on the right represents the average of abundance change and standard deviation of different bacteria between the seedling and FBS stages.

### 3.4. Differences in Soil Fungal Abundance between the Seedling and FBS Stages

Of the eleven soil fungi phyla observed in different samples, Ascomycota was most abundant, accounting for 85.06% and 83.87% of fungal communities in the seedling and FBS stages, respectively (Figure 5). In addition, three fungal phyla (Basidiomycota, Mortierellomycota, and an unclassified phylum) were present in the soil at relative abundances >1%. No significant difference in the relative abundances of fungal phyla between the seedling and FBS stages was found using a Wilcoxon rank-sum test with FDR adjustment ($p > 0.05$). At the class level, the relative abundances of Pezizomycetes and Sordariomycetes were more abundant in soils from the seedling stage, whereas Tremellomycetes, Dothideomycetes, and Eurotiomycetes were significantly enriched in soils from the FBS stage ($p < 0.05$, Figure 5b). Moreover, several fungal genera were more abundant in soils from the FBS stage, including Vishniacozyma, Mycosphaerella, Cladosporium, Filobasidium, and Aspergillus ($p < 0.05$, Figure 5c).

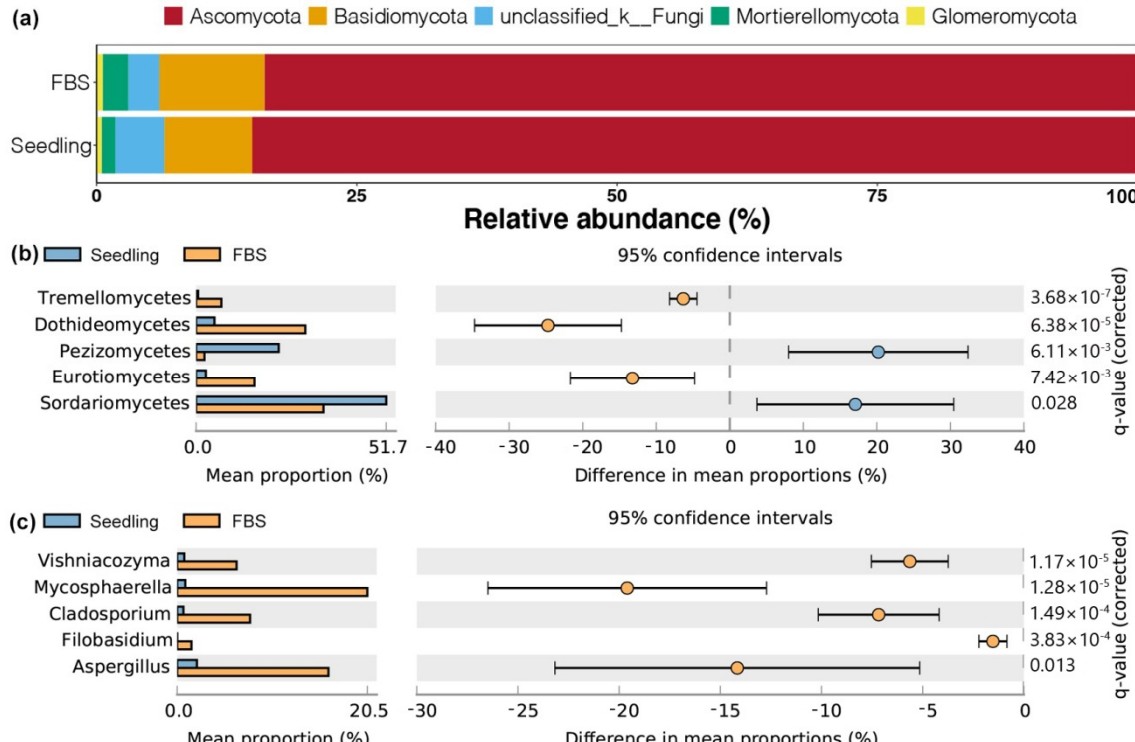

**Figure 5.** The average relative abundance of soil fungal phyla in cotton planting fields at the seedling and FBS stages (**a**). Inter-group difference test of relative abundance of fungi in soils between the seedling and FBS stages at the class (**b**) and genus (**c**) levels, respectively.

### 3.5. Co-Occurrence Network Analysis of Soil Bacterial and Fungal Communities

Molecular ecological network analyses were conducted to reveal the bacterial and fungal interactions in soils at the seedling and FBS stages (Figure 6). The results were used to reveal network complexity and indicated that the soil bacterial community was more complex than the fungal community, though no obvious differences were found between the two communities at different growth stages (Table 1). In addition, the numbers of total nodes and edges for bacteria and fungi between the seedling and FBS stages did not significantly change (Table 1). Interestingly, the number of negative edges for soil bacterial communities in the seedling stage was twice as high as that in the FBS stage (Table 1), suggesting that soil bacterial communities in the seedling stage were more stable.

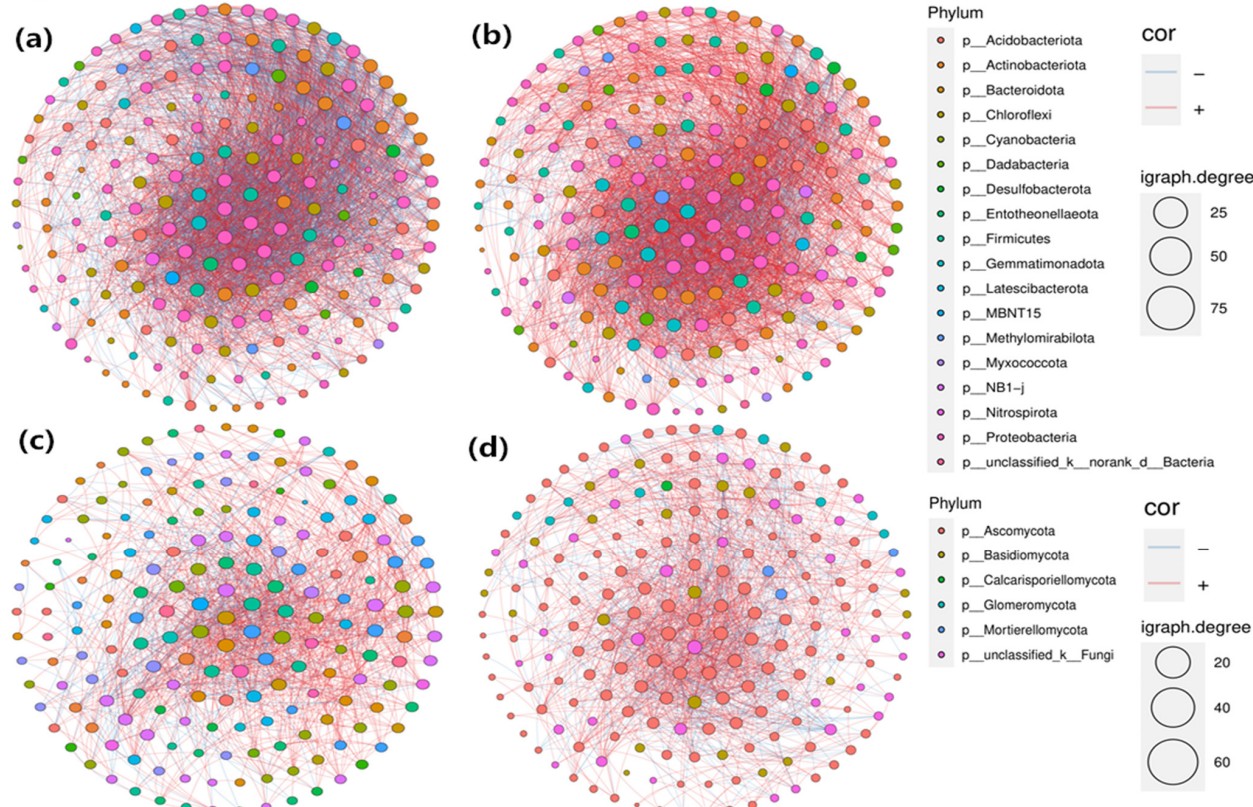

**Figure 6.** Network analysis showing the interactions in soil bacterial and fungal communities in cotton planting fields at the seedling (**a**,**c**) and FBS stages (**b**,**d**).

**Table 1.** Topological parameters of bacterial and fungal co-occurrence networks at the seedling and FBS stages, respectively.

|  | Bacteria-Seedling | Fungi-Seedling | Bacteria-FBS | Fungi-FBS |
|---|---|---|---|---|
| Num edges | 4971 | 2106 | 4832 | 2158 |
| Pos edges | 2668 | 1565 | 3610 | 1544 |
| Neg edges | 2303 | 541 | 1222 | 614 |
| Num nodes | 199 | 191 | 199 | 199 |
| Connectance | 0.252 | 0.116 | 0.245 | 0.110 |
| Average degree | 49.960 | 22.052 | 48.563 | 21.688 |
| Average path length | 1.851 | 2.205 | 1.828 | 2.234 |
| Diameter | 1.890 | 2.333 | 1.919 | 2.292 |
| Clustering coefficient | 0.610 | 0.423 | 0.581 | 0.421 |
| Centralization degree | 0.308 | 0.210 | 0.250 | 0.199 |
| Centralization betweenness | 0.010 | 0.025 | 0.013 | 0.018 |
| Centralization closeness | 0.282 | 0.230 | 0.211 | 0.219 |

### 3.6. Microbial Communities in Relation to Soil Properties and Plant Growth

Mantel tests were used to reveal the relationship between the soil microbial community at the OTU level at different growth stages with respect to soil properties and growth indices (Table 2). For soil bacterial communities, significant correlations were found with the concentrations of salinity, ammonia, nitrate, and phosphate in the seedling stage and the concentrations of salinity, nitrate, moisture, and phosphate in the FBS stage. In contrast, nitrate and phosphate were significantly correlated with soil fungal communities at the seedling stage, and only ammonia was significantly correlated with soil fungal communities at the FBS stage. For the growth indices, soil bacterial communities were significantly correlated with all indices in both the seedling and FBS stages, whereas

a significant correlation was only found between the soil fungal communities and the emergence rate at the seedling stage. These results suggested that soil bacteria, rather than fungal community composition, were associated with cotton growth.

**Table 2.** Mantel tests of soil bacterial and fungal communities with the soil properties and cotton growth indices in the seedling and FBS stages, respectively.

|  | Bacteria–Seedling | | Bacterial–FBS | | Fungi–Seedling | | Fungi–FBS | |
|---|---|---|---|---|---|---|---|---|
|  | r | *p* | r | *p* | r | *p* | r | *p* |
| Salinity | 0.287 | 0.003 | 0.203 | 0.039 | 0.058 | 0.312 | 0.033 | 0.338 |
| Ammonia | 0.205 | 0.031 | 0.161 | 0.096 | 0.087 | 0.237 | 0.261 | 0.017 |
| Nitrate | 0.579 | 0.001 | 0.234 | 0.008 | 0.255 | 0.008 | 0.029 | 0.309 |
| Moisture | 0.085 | 0.149 | 0.258 | 0.036 | 0.024 | 0.380 | −0.103 | 0.778 |
| Phosphate | 0.158 | 0.032 | 0.194 | 0.032 | 0.185 | 0.026 | −0.028 | 0.587 |
| pH | −0.084 | 0.795 | 0.167 | 0.105 | −0.102 | 0.820 | 0.028 | 0.382 |
| SOM | 0.090 | 0.154 | 0.243 | 0.083 | −0.054 | 0.694 | 0.218 | 0.074 |
| Plant height | 0.238 | 0.013 | 0.424 | 0.002 | −0.037 | 0.614 | 0.065 | 0.256 |
| Leaf area | 0.223 | 0.021 | 0.389 | 0.005 | −0.039 | 0.615 | 0.102 | 0.205 |
| Emergence rate/Yield | 0.398 | 0.004 | 0.419 | 0.002 | 0.183 | 0.048 | 0.089 | 0.181 |

To explore the associations between soil microbes and soil properties and cotton growth, correlation analyses were performed on the main bacterial and fungal genera (Figure 7). In the seedling stage, several bacterial and fungal genera were significantly correlated with the emergence rate, among which bacterial genera (Pseudarthrobacter and Thiobacillus) were positively correlated and fungal genera (Cephalotrichum, Chaetomium, and Fusarium) were negatively correlated (Spearman correlation, $p < 0.05$, Figure 6a). In contrast, these bacterial and fungal genera were significantly negatively and positively correlated with soil salinity and nitrate concentrations, respectively (Spearman correlation, $p < 0.05$, Figure 7a). In addition, Planococcus and Fusarium were found to have positive and negative correlations with plant height and leaf area, respectively (Spearman correlation, $p < 0.05$, Figure 7a). Moreover, the fungal genera Coprinellus and Fusarium were negatively correlated with ammonia concentration and plant height, respectively (Spearman correlation, $p < 0.05$, Figure 7a). These findings showed that the associations between soil microbes and cotton growth were more powerful in the seedling stage. More importantly, bacteria and fungi had opposite effects on the emergence rate of cotton, which could be regulated by soil salinity and nitrate concentration.

In the FBS stage, only Fictibacillus (bacteria) was significantly positively correlated with leaf area and yield, and Pseudarthrobacter (bacteria) and Chaetomium (fungi) were significantly negatively correlated with soil salinity and pH, respectively (Spearman correlation, $p < 0.05$, Figure 7b). These findings, combined with the Mantel test results, confirmed that soil properties had stronger effects on soil microbial communities in the seedling stage than in the FBS stage.

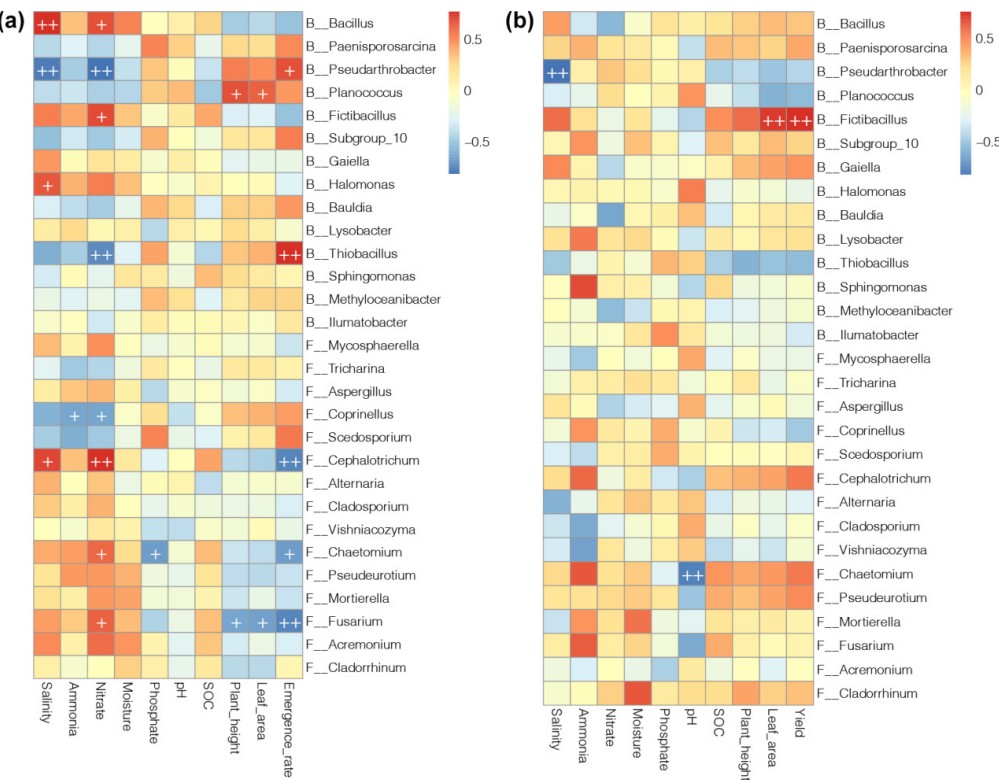

**Figure 7.** Heatmaps of correlations between the abundances of main bacterial and fungal genera with the soil properties and cotton growth indices in the seedling (**a**) and FBS (**b**) stages. The Spearman correlation coefficient is displayed by the color of each cell in the heatmap. A significant correlation was confirmed if the *p*-value with Bonferroni adjustment was less than 0.05 (+) and 0.01 (++).

## 4. Discussion

### 4.1. Variations in Soil Microbial Communities during Crop Development

Determining the dynamics of soil microbial communities during crop growth is important for maintaining ecosystem function and the sustainability of cropping systems [37]. Changes in crop root exudates at different growth stages can also significantly affect soil microbial communities [38,39] and their ability to regulate the growth promotion and disease suppression of crops [40]. Several studies have observed significant changes in soil microbial communities in different crop growth stages [41–43], while changes in soil properties by growth stage can directly or indirectly affect these communities [44]. A previous study documented an obvious succession of soil bacterial communities at key rice growth stages in paddy soil from a rice-wheat cropping system, where pH, OM, and AK appeared to be key factors responsible for the microbial community changes [45]. We found that the biomass, diversity, and composition of soil microbial communities were significantly different between the seedling and FBS stages (Figures 2 and 3). Multiple soil properties were correlated with soil microbial communities (Table 2 and Figure 7). The changes in soil microbial communities among different growth stages were likely induced by variations in soil properties due to various root secretions and nutrient uptake by the plants [46].

### 4.2. Different Responses of Bacterial and Fungal Communities during Cotton Growth

The biomass and richness of both communities were higher in the seedling stage than in the FBS stage (Figure 2). In general, nutrient availability predominantly controls soil microbes [47]. The higher biomass and richness of microbial communities in the seedling stage could be due to the higher availability of nutrients in soils at the onset of the growing season. Molecular ecology network analysis has been widely used to explore the interactions between species in communities [48], improving our understanding of

dynamic changes in microbial niches [49]. The complexity of both bacterial and fungal ecological networks did not clearly change between the seedling and FBS stages. In contrast, the stability of the soil bacterial communities decreased during the FBS stage (Table 1), consistent with the lower richness and diversity of bacterial communities in the FBS stage, indicating that some specific bacteria could be enriched in the latter.

Variations in the abundance of different microbes in cotton fields between different growth stages can reflect their functions or adaptation strategies for cotton growth. For example, the relative abundance of Firmicutes was significantly higher in the FBS stage than in the seedling stage (Figure 5). It can form spores, resting stages that are inactive, strongly dehydrated, and highly resistant to environmental stresses [7]. As these do not exhibit active metabolism and require no energy, they can accumulate in arid soils [50]. In contrast, the relative abundance of three bacterial phyla (Proteobacteria, Nitrospirota, and Entotheonellaeota) was higher at the seedling stage (Figure 4). Proteobacteria are classified as fast-growing copiotrophic bacteria because they prefer nutrient-enriched conditions [51] and have been shown to play an important role in fixing atmospheric N2 to ammonia and providing it to the host plant [52,53]. Nitrospirota as nitrifiers have been demonstrated to exist in various natural ecosystems, with important contributions to ammonia and nitrite oxidation [54]. These findings indicate that nutrient cycling could be more powerful in cotton-planted soils at the seedling stage than at the FBS stage.

*4.3. Stronger Bacterial Association with Cotton Growth*

We concluded that there was a significant correlation between cotton growth and shifts in the bacterial community but not with shifts in the fungal community (Table 2). Similar results were also reported in paddy soils with short-term organic amendments [55] and greenhouse-based high-input vegetable soils [56]. The number of bacteria in soils was significantly higher than that of fungi, leading to a higher contribution of microbial functions [56]. In addition, bacteria are smaller than fungi, suggesting that they are more likely to disperse widely [57] and therefore have a much shorter turnover and rapid response to environmental changes [58]. Moreover, the higher complexity of the bacterial ecological network detected in this study (Figure 6) suggested that bacterial communities represented more powerful metabolic adaptations and occupied more microbial functions [59]. Consequently, bacteria in cotton-planted soils contributed more to microbial function than fungi.

The key question addressed in this study was the primary factor driving the succession of soil bacterial communities and the subsequent impacts on cotton growth. In general agreement with previous studies conducted in various agricultural soils [60], soil nitrates were the dominant force driving variations in the composition of the bacterial community (Figure 7a). In previous studies, nitrogen was commonly regarded as a required element for microbial decomposition of organic materials through the promotion of specific microbial growth that affected community composition [61]. However, in our study, higher nitrate concentrations in the soil resulted in a decrease in beneficial bacteria. One possible explanation for this result is that the experimental soil was saline-alkali, and the rational application of nitrogen fertilizer reduced the negative effects of salt on crop growth and yield [62,63]. However, the extreme nitrate content could inhibit bacterial activity and limit crop growth. Thus, maintaining soil nitrate content in a healthy state is important for promoting cotton growth [64].

## 5. Conclusions

The microbial biomass, diversity, and community structure changed over time, resulting in decreasing microbial biomass and richness in the FBS stage compared to the seedling stage. The stability of bacterial communities was reduced in the FBS stage, while there was no difference in either community's complexity between the seedling and FBS stages. The Mantel test further confirmed that the bacterial community was strongly correlated with soil properties and cotton growth indices, while the fungal community had a very

limited contribution to cotton growth. The changes in soil bacteria that correlated with cotton growth were mainly driven by soil nitrate content. These results provide a novel understanding of the relationships between soil microbial communities and cotton growth at key growth stages.

**Author Contributions:** D.L., M.L., H.Q. and X.Z. designed this study; Y.Y. and Y.Z. performed the field investigation and collected the data; G.T. conducted the language editing. All authors have read and agreed to the published version of the manuscript.

**Funding:** This study was supported by the National Natural Science Foundation of China (51790533 (a major project), 51709266); and the Agricultural Science and Technology Innovation Program (ASTIP) of Chinese Academy of Agricultural Sciences.

**Institutional Review Board Statement:** Not applicable.

**Informed Consent Statement:** Not applicable.

**Data Availability Statement:** Most of the collected data are contained in the tables and figures in the manuscript.

**Conflicts of Interest:** The authors declare no conflict of interest.

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
