# Peer review of "Differing Roles of Bacterial and Fungal Communities in Cotton Fields by Growth Stage"

_agronomy, doi:10.3390/agronomy12030657_

Round 1

Reviewer 1 Report

They analyzed the roles of different communities of bacteria and fungi found in cotton fields depending on the stage of its growth. This research is important for science but also for people involved in cotton cultivation, where china and India are world tycoons. The tests were carried out in accordance with the requirements for scientific research. 

Niestey the author could not avoid minor mistakes:
1. There are no items from poems 48, 53, 64, 65, 189, 355 in the list of literature.
2. There are in the list of literature, but there is no reference in the text of the items: 3, 4, 6, 9, 18, 28, 29, 31, 33, 34, 39, 41, 42, 48, 56, 58, 59.
3. There should be no space between the value and the degree of Celcius, and unfortunately it is in the rows: 131, 132, 174, 175.
4. No superscript for units in rows: 127, 145, 146, 148, 149.

Author Response

Dear Editors and Reviewer:

The authors wish to express their gratitude to the Reviewers and the Editors for their valuable comments.

Those comments about the paper submitted to Agronomy-1617077 are valuable and helpful for revising and improving the paper. The manuscript has been carefully revised according to the comments. Some unclear presentations were rewritten or corrected of the revised manuscript. These changes were highlighted in red in this response letter. All of the reviewer’s comments and the responses are listed as follows.

Reviewer 1#: They analyzed the roles of different communities of bacteria and fungi found in cotton fields depending on the stage of its growth. This research is important for science but also for people involved in cotton cultivation, where china and India are world tycoons. The tests were carried out in accordance with the requirements for scientific research.

Response: Thanks for your confirmation!

Niestey the author could not avoid minor mistakes:

  1. There are no items from poems 48, 53, 64, 65, 189, 355 in the list of literature.

Response 1: We have corrected these references in the revised manuscript. Please check Line 452, 551, 499, 555 of the revised Version.

  1. There are in the list of literature, but there is no reference in the text of the items: 3, 4, 6, 9, 18, 28, 29, 31, 33, 34, 39, 41, 42, 48, 56, 58, 59.

Response 2: After careful checkup, references 3, 19, 33, and 39 were typos, and we have corrected them. Other references have been cited in the text. Please check Line 173, 391, 357, 405, 394, 173, 190, 396, 379, 375, 405, 378 of the revised Version.

  1. There should be no space between the value and the degree of Celcius, and unfortunately it is in the rows: 131, 132, 174, 175.

Response 3: We have corrected it in the revised manuscript. Please check Line 127, 131, 132, 175, 176 of the revised Version.

  1. No superscript for units in rows: 127, 145, 146, 148, 149.

Response 4: We have corrected it in the revised manuscript. Please check Line 127, 145, 146, 148, 149, 153 of the revised Version.

Reviewer 2 Report

This is a review of the article entitled “Differing roles of bacterial and fungal communities in cotton 2 fields by growth stage” submitted to Agronomy by MDPI. In this study, the authors analyzed the soil properties and bacterial compositions of cotton field soils at different developmental stages, namely seedling and flowering/boll-setting (FBS) to determine the influence of these disparate environments on the soil. Through the use of subsurface drainage treatments of the soils and high-throughput sequencing of 16S rRNA and internal transcribed spacer genes, the authors have shown that the bacterial community in the soil had strong influences on the soil properties and cotton growth, driven by nitrate content. 
Well tested studies and thorough discussion demonstrate how the microbial biomass, diversity, and community structure change over time during cotton development and can provide important insight into environmental adaptations and growth treatments. Below are some minor comments for the authors to consider for improving the clarity of this article.

Minor comment: 

Figure 4a and 4b. Hard to read labels of pie graph, either make larger or create a separate legend.

Author Response

Dear Editors and Reviewer:

The authors wish to express their gratitude to the Reviewers and the Editors for their valuable comments.

Those comments about the paper submitted to Agronomy-1617077 are valuable and helpful for revising and improving the paper. The manuscript has been carefully revised according to the comments. Some unclear presentations were rewritten or corrected of the revised manuscript. These changes were highlighted in red in this response letter. All of the reviewer’s comments and the responses are listed as follows.

Reviewer 2#: This is a review of the article entitled “Differing roles of bacterial and fungal communities in cotton 2 fields by growth stage” submitted to Agronomy by MDPI. In this study, the authors analyzed the soil properties and bacterial compositions of cotton field soils at different developmental stages, namely seedling and flowering/boll-setting (FBS) to determine the influence of these disparate environments on the soil. Through the use of subsurface drainage treatments of the soils and high-throughput sequencing of 16S rRNA and internal transcribed spacer genes, the authors have shown that the bacterial community in the soil had strong influences on the soil properties and cotton growth, driven by nitrate content. 

Well tested studies and thorough discussion demonstrate how the microbial biomass, diversity, and community structure change over time during cotton development and can provide important insight into environmental adaptations and growth treatments. Below are some minor comments for the authors to consider for improving the clarity of this article.

Response: Thanks for your confirmation!

Minor comment:

Figure 4a and b. Hard to read labels of pie graph, either make larger or create a separate legend.

Response: we have revised figure 4 and 5. Please check Line 267 and 288.

Reviewer 3 Report

As mentioned in the article and particularly in the Introduction "Many bacteria in soil environments depend on fungal hydrolysis of materials to form primary substrates" it is important to get information of all the microbes including bacteria as well as fungi/oomycetes. Therefore I suggest performing extensive metagenomics for the collected soil samples.

"We performed high-throughput sequencing-based on 16S rRNA and internal transcribed spacer (ITS) genes to identify the bacterial and fungal communities" - here 16S and ITS is not sufficient to get bacterial and fungal communities.

Another major concern is the narrow range of sites considered for the soil sample collections. The authors have shown a map of the entire china to indicate the location of the experiment that gives a false sense. 

I suggest changing figure 1

Please revisit the statistical analysis. In Figure 2, only Shannon looks significant. 

Figure 4 legends need to be well elaborated 

Author Response

Dear Editors and Reviewer:

The authors wish to express their gratitude to the Reviewers and the Editors for their valuable comments.

Those comments about the paper submitted to Agronomy-1617077 are valuable and helpful for revising and improving the paper. The manuscript has been carefully revised according to the comments. Some unclear presentations were rewritten or corrected of the revised manuscript. These changes were highlighted in red in this response letter. All of the reviewer’s comments and the responses are listed as follows.

Reviewer 3#: As mentioned in the article and particularly in the Introduction "Many bacteria in soil environments depend on fungal hydrolysis of materials to form primary substrates" it is important to get information of all the microbes including bacteria as well as fungi/oomycetes. Therefore, I suggest performing extensive metagenomics for the collected soil samples.

Response: This manuscript is not concerned with the interaction of bacteria and fungi, therefore, we did not perform metagenomics to analyze microbial functions. In future studies, we will use metagenomic sequencing to conduct research.

"We performed high-throughput sequencing-based on 16S rRNA and internal transcribed spacer (ITS) genes to identify the bacterial and fungal communities" - here 16S and ITS is not sufficient to get bacterial and fungal communities.

Response: Thank you very much for your valuable comments. We can indeed obtain more information about soil bacteria and fungi by metagenomics. However, we use high throughput methods to obtain the effects of soil environmental factors on microbial communities, and the high throughput sequencing is a commonly used technique at present, which also can be used to obtain the community structure of soil bacteria and fungi. The references are as follows.

In the follow-up experiment, we will use metagenomic sequencing to understand the interaction between soil bacteria and fungi.

  1. Lucas, J. M., Mc Bride, S. G., & Strickland, M. S. (2020). Trophic level mediates soil microbial community composition and function. Soil Biology and Biochemistry, 143, 107756.
  2. Ren, C., Zhang, W., Zhong, Z., Han, X., Yang, G., Feng, Y., & Ren, G. (2018). Differential responses of soil microbial biomass, diversity, and compositions to altitudinal gradients depend on plant and soil characteristics. Science of the total environment, 610, 750–758.

Another major concern is the narrow range of sites considered for the soil sample collections. The authors have shown a map of the entire china to indicate the location of the experiment that gives a false sense.

Response: To avoid readers' misunderstanding of the experiment site and sampling points, we have revised figure 1 and added the label of the experiment site. Please check Line 109.

I suggest changing figure 1

Response: We have corrected it in the revised manuscript. Please check Line 109.

Please revisit the statistical analysis. In Figure 2, only Shannon looks significant.

Response: We have revisited the statistical analysis results, and we confirmed that the figure was correct.

Figure 4 legends need to be well elaborated.

Response: We have detailed this figure legend. Please check Line 270-274 of the revised Version.